# Seed Predation on Oil-Polluted and Unpolluted *Vachellia* (*Acacia*) Trees in a Hyper-Arid Desert Ecosystem

**DOI:** 10.3390/insects11100665

**Published:** 2020-09-28

**Authors:** Marco Ferrante, Daniella M. Möller, Gabriella M. Möller, Yael Lubin, Michal Segoli

**Affiliations:** 1Mitrani Department of Desert Ecology, Ben-Gurion University of the Negev, Midreshet Ben-Gurion 8499000, Israel; moller.daniella@gmail.com (D.M.M.); gabriellamoller1@gmail.com (G.M.M.); Lubin@bgu.ac.il (Y.L.); msegoli@bgu.ac.il (M.S.); 2CE3C-Centre for Ecology, Evolution and Environmental Changes, Azorean Biodiversity Group, Faculty of Agricultural and Environmental Sciences, University of the Azores, PT-9700-042 Angra do Heroísmo, Portugal

**Keywords:** bruchid, ecological functions, Evrona Nature Reserve, germination, oil spill

## Abstract

**Simple Summary:**

Bruchid beetles are the pests of many plant species worldwide. One or more grubs can develop inside a seed by consuming it and impairing its germination. *Vachellia* trees are important for preserving diverse and healthy arid ecosystems, but they are often threatened by human activities and hostile environmental conditions. Seed predation by bruchid beetles is one of the major causes of the decline in the populations of *Vachellia* trees in Israel. In a hyper-arid desert ecosystem affected by two major oil spills (in 1975 and 2014), we evaluated whether oil pollution increases seed predation rates of the seeds of *Vachellia tortilis* and *V. raddiana*. We recorded remarkably high predation rates for both species, particularly at the ground level, which suggests that conservation measures to reduce repeated infestations on fallen pods may be important to preserve these tree species. However, we found no clear evidence of a negative effect of oil pollution on seed predation, indicating that it did not increase the vulnerability of the seeds to bruchids even in trees affected by the recent oil spill.

**Abstract:**

*Acacia* trees are keystone species in many arid environments, supporting high levels of plant and animal diversity. In Israel, the populations of *Vachellia* (formerly *Acacia*) *tortilis* (Forssk.) and *V. raddiana* (Savi) are declining at an alarming rate. Severe infestations by bruchid beetles (Coleoptera, Chrysomelidae) are among the major causes of seed mortality, but additional environmental stressors can reduce the defence level of the seeds, exacerbating their susceptibility to predators. In a hyper-arid desert ecosystem affected by two major oil spills (in 1975 and 2014), we quantified seed predation rates caused by insect granivores before and after the pods dropped to the ground. We recorded predation rates of up to 84% for both tree species, and higher predation rates at the ground level than in the canopy, suggesting that repeated infestations occur. These results reinforce the call to protect the populations of large ungulates such as gazelles, which kill the bruchids by feeding upon the pods, and promote seed germination and dispersion. We found no clear evidence of a negative effect of the oil spill on seed predation, indicating that oil pollution did not increase the vulnerability of the seeds to granivores even in trees affected by the recent oil spill.

## 1. Introduction

Keystone species are species whose presence is crucial to maintain ecological communities [1]. The loss of a keystone species is often followed by a dramatic decline in diversity, which makes keystone species particularly relevant for biodiversity conservation. Tree species enduring hostile environments can act as keystone species, having disproportionate beneficial effects through an increase in productivity and enhancement of favourable microclimatic conditions. *Acacia* trees in arid environments are often such keystone species [2]. *Acacia sensu lato* includes four genera of flowering plants native to semi-arid and arid environments of all continents except Europe [3]. *Acacia* trees generate spatial heterogeneity and shaded areas for plants and animals [4], increase the content of nitrogen in the soil [2], and provide food resources for many invertebrates [5,6] and vertebrates [7] enhancing plant and animal diversity. Acacias can tolerate extreme abiotic conditions [8], being able to survive in some of the most challenging regions of the world.

The Arava Valley in Israel extends approximately from the Dead Sea to the Red Sea, and is characterised by extremely low precipitation (<25 mm annual rainfall) and a high average temperature of around 39 °C in August, the hottest month [9]. The only tree species that survive in this environment are *Vachellia* (formerly *Acacia*) *tortilis* (Forssk.), *V. raddiana* (Savi), and *V. gerrardii* (Benth.) subsp. *negevensis*, with the latter being restricted to high elevations. *Vachellia tortilis* and *V. raddiana* are generally considered to be two subspecies of *V. tortilis* (*V. tortilis* subsp. *tortilis* and *V. tortilis* subsp. *raddiana*), but in Israel, morphological, genetic, and ecological differences between them suggest that they can be regarded as separate species [10,11].

The populations of *Vachellia tortilis* and *V. raddiana* in the Arava, as well as in other parts of the country, are suffering high mortality and declining at an alarming rate [12,13]. The main causes are likely anthropogenic activities, particularly habitat destruction due to road construction, which also alters the natural water flow and modifies the riverbeds where the trees naturally grow. Water extraction for agriculture may also be relevant in exceptionally dry years [14]. Additional factors that might contribute to the decline of *Vachellia* populations include: overgrazing by wild and domestic animals [15], parasitic plants such as *Loranthus acaciae* Zucc. [16], the absence of large mammals that enhance seed germination through pod ingestion [14,17], and high seed predation rates [9]. 

Infestations by bruchid beetles (Coleoptera: Chrysomelidae)—the main acacia seed predators—can be remarkably severe. In the Arava, more than 95% of *Vachellia tortilis* and *V. raddiana* seeds that are not removed by ungulates may be damaged by bruchids [9]. These high seed predation records may be a consequence of trees being in poor physiological conditions because of hydric stress [18]. Stressed *Vachellia* trees can lower the production of nonprotein amino acids that defend their seeds from predators [19]. The poor physiological condition of trees can also be caused or exacerbated by pollutants. Crude oil is one of the major pollutants in the environment [20] and many oil spills occur in deserts [21]. Oil pollution can reduce photosynthetic rates [22] and alter the natural concentrations of proteins, free amino acids, phenols, and sugars in plants [23]. Moreover, the production of defensive secondary metabolites, as well as the assimilation of nutrients and water, may be impaired by oil hydrocarbons [24].

Two major oil spills occurred in the Arava within the past 50 years. Oil from these spills has been shown to accumulate in the soil, negatively affecting *Vachellia* seedling survival and growth [25], and dramatically reducing the recruitment of young trees [26]. However, the interaction between oil pollution and biotic stresses such as seed predation on *Vachellia* trees has not yet been evaluated. The aims of this study were to explore the predation rates caused by invertebrate seed predators before and after the pods dropped to the ground, and to evaluate whether oil pollution increased the natural seed predation rates. To this end, we quantified the seed predation rates on *V. tortilis* and *V. raddiana* in the hyper-arid Arava desert in areas affected by past (1975) and recent (2014) oil spills. We hypothesised that (1) seed predation will be higher on oil-polluted than on unpolluted trees, in accordance with the environmental stress hypothesis [27], which predicts environmental stressors to increase plant vulnerability to herbivores. Additionally, we hypothesised that (2) seed predation rates on the fallen pods will be higher than on the hanging pods, as hanging pods can only be attacked by bruchids on the tree, while fallen pods likely suffer damage by bruchids active both on trees and on the ground. Moreover, fallen pods that are not quickly consumed by large ungulates can be infested by bruchids repeatedly. Finally, we hypothesised that (3) the percentage of seeds consumed in a pod will be negatively related to the number of seeds in the pod. Due to the dilution effect, the chance that at least some seeds will remain undamaged is likely to be greater in larger than in smaller pods. 

## 2. Materials and Methods 

### 2.1. Study Area

Our study area was the Evrona Nature Reserve (29°40′38.76″ N, 35°0′55.12″ E), a hyper-arid desert ecosystem in the Arava valley, which is part of the Afro-Syrian rift valley in southern Israel. The only tree species in this area are *V. tortilis* and *V. raddiana. Vachellia tortilis* flowers once in early summer, while *V. raddiana* has a longer flowering period with two main flowering episodes, first in late spring and a second time in early autumn. A model with seed number as the response and tree species as factor indicated that *V. raddiana* pods contain significantly (ANOVA, *p* < 0.001) more seeds (mean = 9.0, SD = 3.9, n = 251) than those of *V. tortilis* (mean = 6.2, SD = 3.3, n = 557, this study). 

We investigated two study sites affected by two oil spills in 1975 and 2014, respectively (hereafter “1975-site” and “2014-site”). During the first oil spill, around 10,000 m^3^ of crude oil leaked into the reserve, and this was not followed by any remediation attempt [25,26]. The more recent oil spill deposited 5000 m^3^ of oil; part of the contaminated soil was removed and the remaining was tilled in order to aerate the soil. We selected 30 trees: 10 *V. tortilis* and 10 *V. raddiana* trees in the 2014-site, and 10 *V. tortilis* trees in the 1975-site, as in this site *V. raddiana* trees were scarce. Half of the selected trees were, at the time of the spill, within 2 m of the trajectory of the oil flow (hereafter “polluted”), while the other half were at ≥10 m from the oil flow (hereafter “unpolluted”). Evidence of persistent negative effects of the oil on the bacterial and plant communities, even years after these major oil spills occurred, are provided by Girsowicz et al. [28], Nothers et al. [26], and Tran et al. [29]. Differences in physical properties between oil-contaminated and noncontaminated soil are provided by Gordon et al. [30].

Between June–October 2019, mature intact pods were hand-collected from the trees and from the ground (Table 1) to quantify predation rates in and under oil-polluted and unpolluted trees, except for one of the selected unpolluted *V. tortilis* trees, where we found no mature hanging pods at the time of the collection. Pods were placed individually into Petri dishes at 23 °C and their seeds were inspected approximately biweekly for signs of predation. A round hole in the dry seed indicates that an adult beetle has emerged after the larva successfully developed inside the seed. A seed was considered predated if it had at least one exit hole. We collected emerging beetles, thereby preventing a secondary infestation of the seeds, as well as any other arthropods that emerged from the seed.

### 2.2. Statistical Analysis

For each tree species, we used a generalised linear mixed model (GLMM) with binomial distribution and logit-link, and the predation status (yes/no) of each individual seed as the response. The model for *V. tortilis* included the habitat type (on the tree vs. on the ground), the tree status (oil-polluted vs. unpolluted trees), site (1975 vs. 2014), and the interaction between tree status and site as fixed factors, and tree ID and pod ID as random factors to consider the nested structure of the experimental design. The model for *V. raddiana* was identical to the model above, except that it did not include the site, as this tree species was only sampled in the 2014 site. Model selection was conducted comparing Akaike Information Criteria values [31]. We used the linear regression to test whether the percentage of attacked seeds decreased with an increasing number of seeds in a pod. Despite the damage, partially consumed seeds may occasionally germinate; therefore, we counted the number of holes per seed in fallen pods as an indication for seed viability. Compared to hanging pods, fallen pods are exposed to two habitats (the tree and the ground) and are therefore more likely to suffer secondary infestation events that will produce multiple holes per seed. We used a generalised linear model with Gaussian distribution to evaluate whether mean predation rates on hanging pods per tree differ from the corrected mean ground predation rates for the same tree. The correction was conducted by subtracting the mean predation rate on hanging pods from the mean ground predation rate to account for the inevitable exposure of fallen pods to predators both on the tree and on the ground. We excluded one tree from this analysis for which the corrected mean ground predation rate was a negative value, which was not ecologically meaningful (i.e., predation rate cannot decrease after a pod has fallen). All statistical analyses were performed using the R Software [32] and the *lme4* package [33].

## 3. Results

Overall, we assessed 5717 seeds: 1651 and 607 seeds from hanging pods and fallen pods, respectively, for *V. raddiana*; 1962 and 1497 seeds from hanging and fallen pods, respectively, for *V. tortilis*. Bruchid beetles (n = 1078 altogether) were the most abundant seed predators emerging from the pods. *Bruchidius raddianae* (Anton and Delobel 2003) was the most common species collected (>80% of the trees examined), but *B. buettikeri* Decelle, 1979 and *B. obscuripes* (Gyllenhal, 1839) were occasionally observed. Additionally, 19 grass moth larvae (Lepidoptera, Pyralidae) were also found, and we recorded 24 parasitoid wasps emerging from 23 pods. Most of the parasitoids (n = 22) were collected from *V. tortilis* pods. 

Seed predation did not significantly differ between oil-polluted and unpolluted trees for both *V. tortilis* (mean ± SD, 55.3 ± 13.8% vs. 48.8 ± 18.6%, n = 10 for both, for oil-polluted and unpolluted trees, respectively; GLMM, *p* = 0.554) and *V. raddiana* (33.9 ± 10.8% vs. 43.2 ± 24.4%, n = 5 for both, for oil-polluted and unpolluted trees, respectively; GLMM, *p* = 0.428). For both *V. tortilis* and *V. raddiana*, the best model only identified habitat type (tree or ground) as a significant factor. For both species, seed predation rates were significantly higher on the fallen pods than on the hanging pods (GLMM, *p* < 0.001 for both). The average seed predation rates on *V. tortilis* hanging and fallen pods were 34.6 ± 19.9% (n = 19) and 70.2 ± 15.9% (n = 20), respectively. The average seed predation rates on *V. raddiana* hanging and fallen pods were 22.6 ± 20.2% (n = 10) and 83.9 ± 18.2% (n = 10), respectively (Figure 1). On *V. tortilis,* seed predation rates were higher, although not significantly so, in the 1975-site (53.2 ± 21.8%, n = 10) than in the 2014-site (50.9 ± 9.1%, n = 10).

Contrary to our prediction, we found that the percentage of seeds predated in a pod was unrelated to the total number of seeds in the pod: *V. raddiana*, linear regression, *t* = −0.388, *p* = 0.698, and *V. tortilis*, linear regression, *t* = 0.503, *p* = 0.615. For predated seeds from fallen pods, the average number of holes per seed was 1.3 ± 0.58 (SD = 0.58, n = 1570). More than 80% of the predated seeds had a single hole, 14.5% had two holes, 4.6% had three holes, and 0.2% had four holes. A small fraction of the predated seeds (4.2%) were totally consumed. In fallen pods, we confirmed the previous observation that the same seed may host up to five bruchid individuals [34].

Mean predation rates on hanging pods (28.4 ± 17.8%, n = 28) were significantly lower (GLMM, *p* = 0.003) than corrected mean predation rates on fallen pods (47.2 ± 25.8%, n = 28; Figure 2).

## 4. Discussion

We found no evidence to support hypothesis 1, that oil pollution increases seed predation rates. However, the overall predation rates on seeds of both *Vachellia* species and at both sites was very high (70.2 and 83.9% on fallen pods of *V. tortilis* and *V. raddiana*, respectively). Even if oil pollution may not have a major effect on seed predation, *V. tortilis* and *V. raddiana* seed germination [29] and recruitment [26] are both reduced on oil polluted soils. Oil pollution, in addition to seed predation, can therefore be considered a major threat to the survival of *Vachellia* trees. 

In accordance with hypothesis 2, we observed that seed predation rates were significantly higher on fallen than on hanging pods, increasing from 34.6 to 70.2% for *V. tortilis* and from 22.6 to 83.9% for *V. raddiana*. This pattern remained consistent when comparing the predation rates on hanging pods with corrected predation rates on fallen pods (i.e., after subtracting the mean predation rate on hanging pods). Although very high, these values were lower than those recorded by Rohner and Ward [9] in the northern Arava Valley (96.2% for *V. tortilis*, and 97.6% for *V. raddiana*). These authors, however, might have estimated the predation rates on pods that were exposed to seed predators for longer periods on the ground. In contrast, the predation rates on hanging pods were higher than those recorded by Coe and Coe [35]. We found that the percentage of attacked pods did not depend on the number of seeds in a pod (hypothesis 3 rejected). It is possible that this pattern is a consequence of the remarkably high density of bruchids in our study area. 

There are at least three potential explanations for the high predation rates observed. The first is that predation occurs already on young hanging pods, and that re-infestation events reduce the seed crop further. In our study area, bruchids have been observed all year-round [36], indicating that a small population of these beetles is present in the site even during the winter months, and can start a rapid infestation as soon as *Vachellia* trees produce new pods. Consistent with this idea, we observed up to five exit holes on one seed and many seeds with more than one bruchid larva (ca. 20%), which suggests several independent infestation events. The second explanation is that the high seed predation rates may be a consequence of the low density of the large-bodied vertebrate fauna of Israel feeding on the pods. For example, the critically endangered Arabian Gazelle, *Gazella acaciae*, (Mendelssohn, Groves, and Shalmon), that used to be widespread in the area, is nowadays restricted to a 3.5 km^2^ enclosure in the Yotvata Nature Reserve and numbers 22 individuals, while the population density of the Dorcas Gazelle, *G. dorcas* L., in Evrona Nature Reserve is very low [37]. Ungulates reduce seed predation by controlling the bruchid population, as the beetle dies when ingested with the pods, and by removing part of the fallen pods before new infestations occur [34]. Hence, protecting and sustaining the population of native vertebrate herbivores should be an essential part of conservation strategies aimed at maintaining this ecosystem. The third potential reason for high infestation rates by bruchids may be the low densities of their natural enemies. The low number of parasitoid wasps collected suggests that the impact of parasitism on the bruchid population is probably small, which is in accordance with a study of *V. tortilis* in Botswana [38].

Our observations indicated that at least four bruchid species feed upon *Vachellia* seeds. *Bruchidius raddianae* was the predominant seed predator in our study area. Although the biology of this species is poorly known, the larva has been recorded in pods of several *Vachellia* species [39,40] and can be considered a generalist. According to Anton et al. [41], in Israel, *B. buettikeri* develop on the pods of *V. tortilis*, *V. raddiana*, and *V. gerrardii negevensis*, which are common tree species in southern Israel. Instead, *B. obscuripes* has not been listed for our study area and its host plants are unknown. More studies are needed to determine whether these species should be controlled, and how to protect keystone *Vachellia* trees in this fragile ecosystem.

## 5. Conclusions

Bruchid beetles are a main cause of seed mortality for *Vachellia* trees, and they most likely contribute to the decline of the populations of *V. tortilis* and *V. raddiana* in arid environments in Israel. We found that up to 84% of the seeds were consumed by bruchid beetles, but there was no clear evidence of additional negative effects of oil pollution on seed predation rates. We underline that quantitative measures such as the ones provided by this study are paramount for understanding the great variability observed in bruchid predation, as well as to identify additional factors that may affect seed mortality in *Vachellia* trees in the Arava. 

## Figures and Tables

**Figure 1 insects-11-00665-f001:**
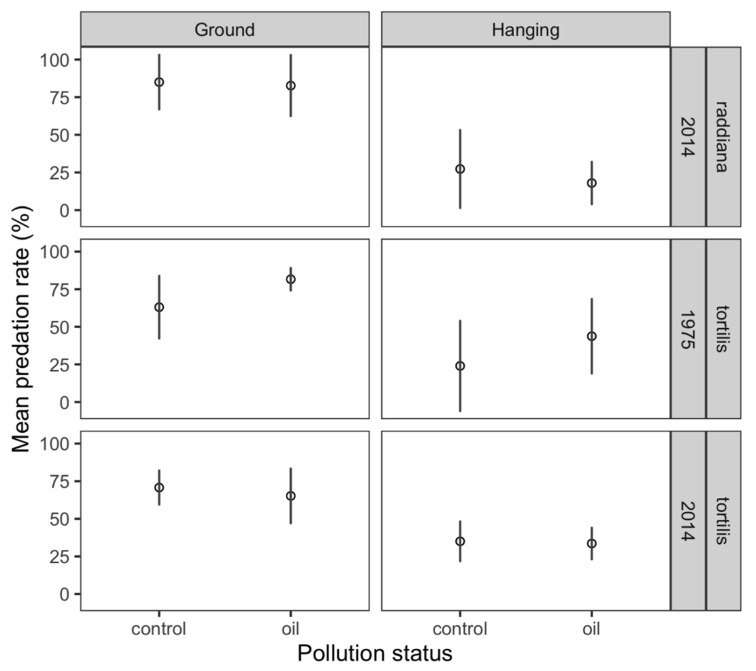
Mean predation rate (% ± SD) on seeds collected from oil-polluted (“oil”) and unpolluted (“control”) *Vachellia tortilis* and *V. raddiana* trees in the two sites of the experiment (1975 and 2014) and from the two habitats (ground and tree) in Evrona Nature Reserve, southern Israel.

**Figure 2 insects-11-00665-f002:**
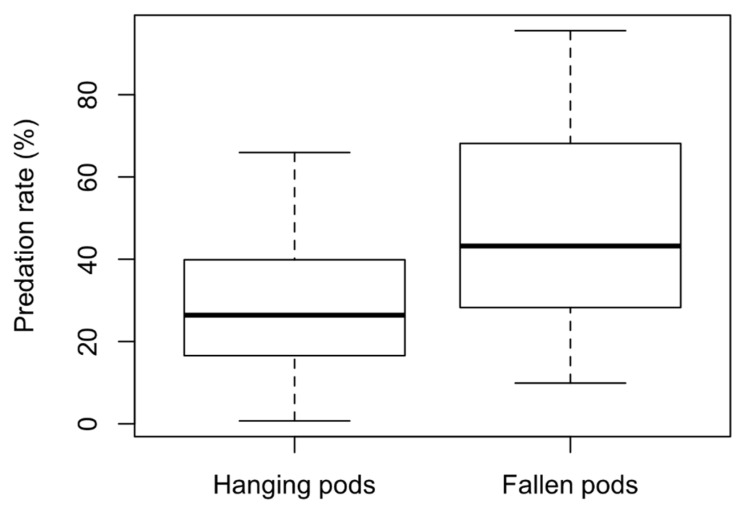
Predation rates (%) on seeds from *Vachellia tortilis* and *V. raddiana* trees hanging and fallen pods in Evrona Nature Reserve, southern Israel. For fallen pods, predation rate was corrected by subtracting the mean predation rate on hanging pods from the mean ground predation rate. The tick line indicates the median, the lower and upper side of the box corresponds, respectively, to the upper and lower quartiles, and the whiskers extend to 1.5 times the interquartile range.

**Table 1 insects-11-00665-t001:** Number of seeds collected per each *Vachellia* tree species, site, tree status, and habitat in Evrona Nature Reserve, southern Israel, during the field season 2019. Inspection dates are the dates on which seeds were examined for the presence of holes and emerging beetles were removed.

*Vachellia* Species	Site, Tree Status	Seeds Collected	Habitat	Collection Date	Inspection Dates
*V. raddiana*	2014, oil-polluted	892	tree	2 June	30 June; 14, 27 July; 11, 27 August; 4, 24 September
	2014, unpolluted	759	tree	2 June	30 June; 14, 27 July; 11, 27 August; 4, 24 September
	2014, oil-polluted	306	ground	2 October	23 October
	2014, unpolluted	301	ground	2 October	23 October
*V. tortilis*	2014, oil-polluted	489	tree	8 August	28 August; 13, 25 September; 14 October
	2014, unpolluted	684	tree	8 August	28 August; 13, 25 September; 14 October
	1975, oil-polluted	396	tree	8 August	28 August; 13, 25 September; 14 October
	1975, unpolluted	393	tree	8 August	28 August; 13, 25 September; 14 October
	2014, oil-polluted	361	ground	2 October	23 October
	2014, unpolluted	402	ground	2 October	23 October
	1975, oil-polluted	399	ground	2 October	23 October
	1975, unpolluted	335	ground	2 October	23 October

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
