# Peer review of "Seed Predation on Oil-Polluted and Unpolluted *Vachellia* (*Acacia*) Trees in a Hyper-Arid Desert Ecosystem"

_insects, 2020, doi:10.3390/insects11100665_

Round 1
Reviewer 1 Report
I would like to thank the authors for their thorough revisions and for their patience with my comments, I am not satisfied with this manuscript!
Reviewer 2 Report
Dear authors,
congratulations on your new version of your MS. I am fully satisfied by both requested changes in the previous version (correction of comparison of hanging pods and pods on ground + identification of bruchids).
I like also some small changes in the Introduction of keystone species.
I have only some comments about possible future studies. You commented about the situation around parasitoids, but you do not know if your parasitoids are a really parasitoids of bruchids. You have there also some moths, and it is necessary to identify them. And you can also think about possible re-introduction of native enemies as biological control (or at least about the study on it).
Best wishes,
PS: In the Acknowledgements, you have there XXXX instead of exact number. Please, do not forget to change it.
This manuscript is a resubmission of an earlier submission. The following is a list of the peer review reports and author responses from that submission.
Round 1
Reviewer 1 Report
In the present study, the authors set out to investigate the influence of oil pollution and bruchid beetle infestation on Vachellia tortilis and V. raddiana seeds. They collected seeds from two sites that had been affected by major oils spills and found very high seed predation rates at both sites and for both Acacia tree species, and that predation rates were higher on the ground than in the canopy. However, they did not find the expected negative influence of oil pollution.
I think this is an interesting study and the manuscript is generally well written. However, I did find a few points that will need to be addressed.
Main points:
1 - Introduction: I suggest starting with something of a 'bigger-picture paragraph' instead of starting with the focus directly on the trees. Maybe start with a short paragraph outlining what keystone species are and why they are so important, in particular to semi-arid and arid environments.
2 - Methods, lines 94-96: Please provide some more information on this model. What were the factors? Since you only mentioned two species, why was this an ANOVA and not a t-test or equivalent?
3 - Methods, line 105: How do you know that this (≥10m) is far enough to be able to call them unpolluted? I assume that some of the contaminants can also move laterally in the soil, e.g., after rainfall or via soil invertebrates? This cut-off will need to be more strongly supported as this otherwise is not really a control and in that case it would not be surprising that you did not find an effect of oil pollution in your analyses.
Specific points:
1 - Introduction, lines 75-83: I suggest numbering your hypotheses here to make it easier to follow them in the discussion.
2 - Introduction, lines 83-87: Please remove this from the introduction as this is foreshadowing of results. Given the overall brevity of the manuscript, this is not needed here.
3 - Methods, lines 124-125: What was the error function for the GLMM?
4 - Methods, lines 136-137: Which library package did you use to conduct the GLMM?
5 - Results, figure 1: Please enlarge the circles indicating the position of the means.
6 - Discussion, lines 171-177: This is largely repetitive with the introduction and, particularly given the short overall manuscript length, should be deleted. Just start directly with your result-interpretation.
7 - Discussion, line 179: This would be much easier to follow if the hypotheses were numbered (please see #1 above). I suggest writing here: 'We found no evidence to support our hypothesis 1 that oil...' and then also linking subsequent sentences to specific, numbered hypotheses (i.e., line 184 and lines 190-191).
Author Response
Dear Editor,
Re. revised submission of MS insects-881610
Thank you for forwarding the reviewer’s comments on our submission and for the fast handling of our manuscript. We are happy to read that the reviewers found our work overall well written and interesting. We have taken into consideration their valuable suggestions and replied to each comment – please find these below. We hope that the manuscript will now be easier to comprehend and suitable for acceptance.
Thank you for your editorial assistance, and we await further from you concerning our manuscript.
Sincerely yours,
Dr. Marco Ferrante
Corresponding author
Reviewer1
In the present study, the authors set out to investigate the influence of oil pollution and bruchid beetle infestation on Vachellia tortilis and V. raddiana seeds. They collected seeds from two sites that had been affected by major oils spills and found very high seed predation rates at both sites and for both Acacia tree species, and that predation rates were higher on the ground than in the canopy. However, they did not find the expected negative influence of oil pollution.
I think this is an interesting study and the manuscript is generally well written. However, I did find a few points that will need to be addressed.
Main points:
1 - Introduction: I suggest starting with something of a 'bigger-picture paragraph' instead of starting with the focus directly on the trees. Maybe start with a short paragraph outlining what keystone species are and why they are so important, in particular to semi-arid and arid environments.
Reply: We followed the recommendation and added the following paragraph: “Keystone species are species whose presence is crucial to maintain ecological communities (Paine 1969). The loss of a keystone species is often followed by a dramatic decline in diversity, which makes keystone species particularly relevant for biodiversity conservation. Tree species enduring hostile environments can act as keystone species, having disproportionate beneficial effects through an increase in productivity and enhancement of favorable microclimatic conditions. Acacia trees in arid environments are often such keynote species.”
2 - Methods, lines 94-96: Please provide some more information on this model. What were the factors? Since you only mentioned two species, why was this an ANOVA and not a t-test or equivalent?
Reply: The model only included seed number as the response and tree species as factor. Both ANOVA and the t-test are suitable to analyze these data and provide comparable results.
3 - Methods, line 105: How do you know that this (≥10m) is far enough to be able to call them unpolluted? I assume that some of the contaminants can also move laterally in the soil, e.g., after rainfall or via soil invertebrates? This cut-off will need to be more strongly supported as this otherwise is not really a control and in that case it would not be surprising that you did not find an effect of oil pollution in your analyses.
Reply: The spill was of heavy, crude oil (long-chain hydrocarbons) and where the oil flowed, it has remained in the upper 30 cm of the substrate. The area is hyper-arid with scarcely any rainfall, such that lateral movement of oil on or just below the surface is unlikely. In any event, the amount would be negligible by comparison with the amounts present in soil in the direct path of the flow.
Specific points:
1 - Introduction, lines 75-83: I suggest numbering your hypotheses here to make it easier to follow them in the discussion.
Reply: Changed as suggested
2 - Introduction, lines 83-87: Please remove this from the introduction as this is foreshadowing of results. Given the overall brevity of the manuscript, this is not needed here.
Reply: Removed as suggested
3 - Methods, lines 124-125: What was the error function for the GLMM?
Reply: The error term of the fixed component of the model followed a binomial distribution (L130-131).
4 - Methods, lines 136-137: Which library package did you use to conduct the GLMM?
Reply: We added the library (lme4) in the text.
5 - Results, figure 1: Please enlarge the circles indicating the position of the means.
Reply: Done.
6 - Discussion, lines 171-177: This is largely repetitive with the introduction and, particularly given the short overall manuscript length, should be deleted. Just start directly with your result-interpretation.
Reply: Removed as suggested.
7 - Discussion, line 179: This would be much easier to follow if the hypotheses were numbered (please see #1 above). I suggest writing here: 'We found no evidence to support our hypothesis 1 that oil...' and then also linking subsequent sentences to specific, numbered hypotheses (i.e., line 184 and lines 190-191).
Reply: Changed as suggested
Reviewer 2 Report
Dear authors,
in my opinion, your MS is well written and you present your results excellently. But still, I feel there missing parts. I think that you can present more than the only easy comparison of the predation rate of oil vs control and hanging vs ground.
Firstly, you should compare the predation rate of hanging versus real predation rate on the ground. You mentioned there, that you should subtract the value of the predation rate of hanging, but later you do not comment, calculate or analyze it. If you know a real predation rate on the ground, then you can compare it with hanging and analyze where is a really higher attack on seeds.
Next, you commented also that one pod has up to five bruchids, but no analysis of it. Do you know the prevalence, the abundance of each predator? It will be useful to present here all data. There can be some differences between hanging vs ground + oil vs control. And of course, what about other predators (you have at least two species of bruchids, moths). There can be some preferences according to plant species, oil vs control, or position of the pod (ground vs hanging). But we have not seen any additional data and results.
Of course, try to identify all predators to species, then it will be very useful (or you can give their photos in the paper).
In my view, if you add all these additional comments by your style, then it will be a perfect paper. Now, it is perfect written MS, but in my view, the value of recent MS has not sufficient value for this journal.
some small comments:
Line 53 - add author for Loranthus acaciae
Lines 83-87 -this is more for results, you present here your results. Here, you can present only aims and hypotheses.
Line 107 - order of authors - should be alphabetically or by years - I have not find your way.
Author Response
Reviewer2
Dear authors,
in my opinion, your MS is well written and you present your results excellently. But still, I feel there missing parts. I think that you can present more than the only easy comparison of the predation rate of oil vs control and hanging vs ground.
Reply: Thank you for considering the ms well written.
Firstly, you should compare the predation rate of hanging versus real predation rate on the ground. You mentioned there, that you should subtract the value of the predation rate of hanging, but later you do not comment, calculate or analyze it. If you know a real predation rate on the ground, then you can compare it with hanging and analyze where is a really higher attack on seeds.
Reply: We cannot know the actual rate of infestation on the ground alone for two reasons. First, the predation rate on the ground includes the predation on the pods when they were still hanging on the tree. Second, the fallen pods collected from the ground and the hanging pods were not from the exact same pool, as in order to assess predation the pods is destroyed). Therefore, we believe that the comparison has to be done between predation rates on hanging pods (only due to bruchids active on the trees) vs. fallen pods (due to bruchids active both on the trees and on the ground level). We have clarified this aspect in the introduction L86-88:
Additionally, we hypothesised that 2) seed predation rates on the fallen pods will be higher than on the hanging pods, as hanging pods could be only attacked by bruchids on the tree, while fallen pods likely suffer damage by bruchids active both on trees and on the ground. Moreover, fallen pods that are not quickly consumed by large ungulates can be infested by bruchids repeatedly.
Next, you commented also that one pod has up to five bruchids, but no analysis of it. Do you know the prevalence, the abundance of each predator? It will be useful to present here all data. There can be some differences between hanging vs ground + oil vs control. And of course, what about other predators (you have at least two species of bruchids, moths). There can be some preferences according to plant species, oil vs control, or position of the pod (ground vs hanging). But we have not seen any additional data and results.
Of course, try to identify all predators to species, then it will be very useful (or you can give their photos in the paper).
Reply: Unfortunately, we were unable to identify with certainty the bruchid species collected, and for this reason we decided to pool all bruchid species in the analysis. We understand that this could have added some value to this work, but our main focus was the quantification of seed predation rates with implications for the sustainability of the tree populations. Additionally, the number of microlepidoptera caterpillars was negligible compared to that of bruchids.
In my view, if you add all these additional comments by your style, then it will be a perfect paper. Now, it is perfect written MS, but in my view, the value of recent MS has not sufficient value for this journal.
some small comments:
Line 53 - add author for Loranthus acaciae
Reply: Done.
Lines 83-87 -this is more for results, you present here your results. Here, you can present only aims and hypotheses.
Reply: Removed as suggested by both reviewers
Line 107 - order of authors - should be alphabetically or by years - I have not find your way.
Reply: Corrected.
Round 2
Reviewer 1 Report
I would like to thank the authors for their careful revision in response to my previous comments. I am happy with most changes made, but have three follow-ups from previous comments, one slightly larger concern and two smaller, specific requests.
Main comment:
Regarding my previous comment on oil-spill vs control sites, I generally accept your reply but still posit that while this is 'logical', you do not provide any scientific data supporting your assertion that therefore these sites >10m away from the spill really are control sites. It would be good to have some soil measurements, for example, that could show how much contamination (if any) is present also at your 'control sites'. You concede in your reply to my original comment that 'In any event, the amount would be negligible by comparison with the amounts present in soil in the direct path of the flow'. Even if that is true, how do you know that even these negligible amounts relative to the higher amount experienced directly in the path of the oil are not enough to have an impact? There could be a very low threshold value which, once surpassed, all responses will be similar. To me, the use of these as control sites is still not fully resolved/justified.
Specific comments:
Material and methods, lines 107-109: Please specify the model structure here as well (and not just in your reply to my previous comment), so that this is also available for your subsequent readers.
Material and Methods, Statistical analysis, lines 138-139: My apologies for the wrong terminology in my previous comment, I meant to enquire about the 'link function' used, not the 'error distribution', which you had already stated in the original version of the manuscript. Please provide the details of the model's link function here as well.
Author Response
REV 1
I would like to thank the authors for their careful revision in response to my previous comments. I am happy with most changes made, but have three follow-ups from previous comments, one slightly larger concern and two smaller, specific requests.
Main comment:
Regarding my previous comment on oil-spill vs control sites, I generally accept your reply but still posit that while this is 'logical', you do not provide any scientific data supporting your assertion that therefore these sites >10m away from the spill really are control sites. It would be good to have some soil measurements, for example, that could show how much contamination (if any) is present also at your 'control sites'. You concede in your reply to my original comment that 'In any event, the amount would be negligible by comparison with the amounts present in soil in the direct path of the flow'. Even if that is true, how do you know that even these negligible amounts relative to the higher amount experienced directly in the path of the oil are not enough to have an impact? There could be a very low threshold value which, once surpassed, all responses will be similar. To me, the use of these as control sites is still not fully resolved/justified.
Reply: We added one reference from our same study area that used a similar configuration of control and treated sites as in our study, and showed a difference between contaminated and non-contaminated soil samples. See L128-129: “Differences in physical properties between oil-contaminated and non-contaminated soil are provided by Gordon et al., [30]”.
Specific comments:
Material and methods, lines 107-109: Please specify the model structure here as well (and not just in your reply to my previous comment), so that this is also available for your subsequent readers.
Reply: We added this information in the text, see L114-115.
Material and Methods, Statistical analysis, lines 138-139: My apologies for the wrong terminology in my previous comment, I meant to enquire about the 'link function' used, not the 'error distribution', which you had already stated in the original version of the manuscript. Please provide the details of the model's link function here as well.
Reply: We added this information in the text, see L154.
REV 2
Dear authors,
unfortunately, I see that you have not accepted my comments (1) to comparison the predation rate of hanging pods versus real predation rate on the ground, and (2) to identify of herbivores from pods.
(1) You comment that the pods are from different pools. Do you mean different trees or area? If you have from a different area than it should be completely rejected because it is impossible to compare it from different areas. If you have one area, but different trees, then it is no problem to compare it.
Reply: The hanging and fallen pods were obtained from the same trees. Yet, we believe that to obtain the real seed predation rates on the ground pods, we would have needed to expose uninfested pods on the ground and collect them after a set period of time. Therefore, while we agreed to repeat the analysis (see below), we refrain from using the word “real”.
I understand that it is not precise to make the real predation on the ground, but now you compare A vs A+B. And of course, there should be a difference. But I would like to know if there is a real difference between attacks on hanging pods versus pods on the ground. Now, we have no comparison. Your results show only that pods on the ground are more attacked but this is absolutely sure.
Reply: we added a model where we test for differences between mean predation rates on hanging pods vs. the corrected mean predation rates on the ground (calculated by subtracting the predation on hanging pods from the ground predation for each tree). These new results are consistent with our conclusions. This analysis has been detailed in the material and methods section at L159-166, in the results L204-205, and in the discussion L231-233. We also included a new figure presenting these results (Figure 2).
(2) Bruchids spp. are a real problem for me. This excellent paper without Bruchids identifications is more "plant ecological paper" than "entomological paper". You wrote that you find 5 species - so you are able to recognize the morphotypes. Then it can be easy to separate it completely. In my view, it is important to see if there are some specialists or generalists, there should be also important data for the biology of these species.
In my view, it is a really perfect written paper, but still, there are missing some data. Despite the high quality of analysing and writing, paper without commented data can not be accepted in Insects in my view.
Best wishes,
JIRISLAV
Reply: We clarified that when we said that the same seed can host up to five bruchids, we meant five individuals (L205).
We have sent our sampled for identification and we now know that most bruchids collected were Bruchidius raddianae (Anton & Delobel 2003). Not much is known about the biology of this species, except that the larva can develop on several Vachellia (formerly Acacia) species (e.g., Derbel et al. 2007; Toma et al. 2017). Bruchidius buettikeri Decelle, 1979 and B. obscuripes (Gyllenhal, 1839) were more rarely observed. According to Anton et al. 1997, in Israel, B. buettikeri develop on Vachellia tortilis, V. raddiana, and V. gerrardii negevensis. Instead, B. obscuripes is not listed for our study area and its host plants are unknown. We added a paragraph about this at the end of the discussion (L262-269).
Anton et al. 1997 "An annotated list of the Bruchidae (Coleoptera) of Israel and adjacent areas." Israel Journal of Entomology 31:59-96.
Derbel et al. 2007 "Life cycle of the coleopter Bruchidius raddianae and the seed predation of the Acacia tortilis Subsp. raddiana in Tunisia." Comptes rendus biologies 330:49-54.
Toma et al. 2017 "First record of Bruchidius raddianae in Italy: infested seeds of Vachellia karroo from Lampedusa island (Coleoptera: Bruchidae; Fabales: Fabaceae)." Fragmenta Entomologica 49:89-91.
Sincerely,
Marco Ferrante
Corresponding author

Reviewer 2 Report
Dear authors,
unfortunately, I see that you have not accepted my comments (1) to comparison the predation rate of hanging pods versus real predation rate on the ground, and (2) to identify of herbivores from pods.
(1) You comment that the pods are from different pools. Do you mean different trees or area? If you have from a different area than it should be completely rejected because it is impossible to compare it from different areas. If you have one area, but different trees, then it is no problem to compare it.
I understand that it is not precise to make the real predation on the ground, but now you compare A vs A+B. And of course, there should be a difference. But I would like to know if there is a real difference between attacks on hanging pods versus pods on the ground. Now, we have no comparison. Your results show only that pods on the ground are more attacked but this is absolutely sure.
(2) Bruchids spp. are a real problem for me. This excellent paper without Bruchids identifications is more "plant ecological paper" than "entomological paper". You wrote that you find 5 species - so you are able to recognize the morphotypes. Then it can be easy to separate it completely. In my view, it is important to see if there are some specialists or generalists, there should be also important data for the biology of these species.
In my view, it is a really perfect written paper, but still, there are missing some data. Despite the high quality of analysing and writing, paper without commented data can not be accepted in Insects in my view.
Best wishes,
JIRISLAV
Author Response

(The authors gave the same response as above.)
